# An Overview of Corneal Transplantation in the Past Decade

**Mutali Musa** [1] **, Marco Zeppieri** [2,*] **, Ehimare S. Enaholo** [3,4] **, Ekele Chukwuyem** [3,4] **and Carlo Salati** [2]

1   Department of Optometry, University of Benin, Benin City 300238, Nigeria
2   Department of Ophthalmology, University Hospital of Udine, 33100 Udine, Italy
3   Centre for Sight Africa, Nkpor, Onitsha 434112, Nigeria
4   Africa Eye Laser Centre, Benin 300001, Nigeria
*   Correspondence: markzeppieri@hotmail.com

**Abstract:** The cornea is a transparent avascular structure located in the front of the eye that refracts light entering the eyes and also serves as a barrier between the outside world and the internal contents of the eye. Like every other body part, the cornea may suffer insult from trauma, infection, and inflammation. In the case of trauma, a prior infection that left a scar, or conditions such as keratoconus that warrant the removal of all or part of the cornea (keratoplasty), it is important to use healthy donor corneal tissues and cells that can replace the damaged cornea. The types of cornea transplant techniques employed currently include: penetrating keratoplasty, endothelial keratoplasty (EK), and artificial cornea transplant. Postoperative failure acutely or after years can result after a cornea transplant and may require a repeat transplant. This minireview briefly examines the various types of corneal transplant methodologies, indications, contraindications, presurgical protocols, sources of cornea transplant material, wound healing after surgery complications, co-morbidities, and the effect of COVID-19 in corneal transplant surgery.

**Keywords:** cornea transplant (CT); penetrating keratoplasty; endothelial keratoplasty (EK); artificial cornea transplant; COVID-19





## 1. Introduction

The cornea is an avascular tissue that is transparent due to the arrangement of its component cells, its avascularity and the metabolic processes its endothelium carries out [1]. It measures 11–12 mm horizontally and 10–11 mm vertically. The cornea is 500–600 μm thick in the center while it gets thicker towards the periphery. The cornea is aspheric and has a refractive index of 1.376. Research has found the average cornea radius to be 7.8 mm, which results in a dioptric power of 43.25 D on the front surface of the cornea, using the keratometer calibration index of 1.3375 [2]. The total dioptric power of a normal human eye is about 58.60 D, and the cornea contributes 74%. This shows the importance of the cornea in vision as it contributes the most to bending light rays. In simpler terms, its curvature influences to a large extent the possibility of refractive error being present. The cornea is the major cause of astigmatism in the human refractive system. A healthy cornea is a trilaminar tissue with the following layers: the epithelium, Bowman's layer (epithelial basement membrane), substantia propria (or stroma), Descemet's membrane (basement membrane of stroma), and endothelium [3].

## 2. Materials and Methods

We searched the Pubmed database using the following search criteria; "corneal transplantation" [MeSH Terms] OR ("corneal" [All Fields] AND "transplantation" [All Fields]) OR "corneal transplantation" [All Fields] OR ("cornea" [All Fields] AND "transplantation" [All Fields]) OR "cornea transplantation" [All Fields]. Given that this study was focused on CT in the last decade, we narrowed the search criteria to 2012 to 2023, resulting in a return of 3021 results. The search words using only "transplant" or "keratoplasty" or other single

words were not considered in our minireview to avoid an overabundance of potentially impertinent studies. This search strategy was limiting and could have potentially and unintentionally excluded opinion leaders in this field of research. A Prisma Table was also generated, showing further stratification steps.

### 2.1. Corneal Transplant—Methodology, Indications, and Contraindications

Corneal transplantation (CT) is indicated via any processes leading to loss of corneal transparency, with neuroretinal integrity and optimal visual potential otherwise preserved [4]. CT is usually required for the management of advanced corneal dystrophies, corneal degenerations, and occluding corneal scars secondary to mechanical, chemical, or thermal injuries, all without comorbid ocular disease conditions to consider (Figure 1).

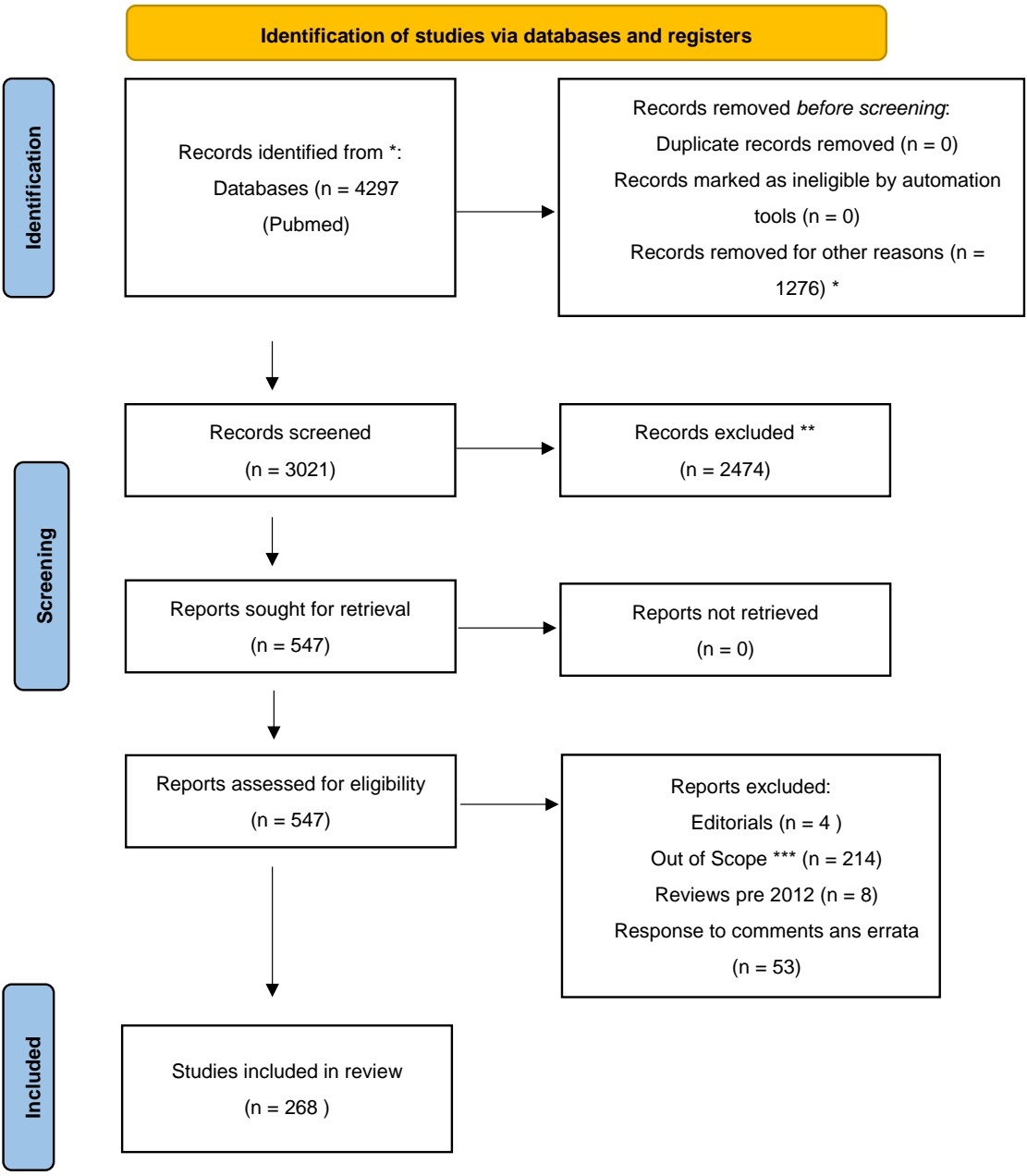

**Figure 1.** Flow Chart depicting a summary of Cornea Transplant. * 1276 records were excluded for being out of range chronologically; ** Records that did not contain the word transplant in their title were removed; *** Records were based on limbal and other body parts transplants.

CTs are among the most common procedure performed in many tertiary eye centers worldwide [5–12]. Various forms of CT are carried out, depending on the laminar depth of scarring or recipient corneal tissue viability. Lamellar keratoplasty is indicated when scarring extends to the corneal epithelium, Bowman's layer, and stroma. This can either be anterior lamellar keratoplasty (ALK) for more superficial graft implantation; or deep lamellar keratoplasty (DLK) [13] when the stroma is equally diseased. The technique of penetrating keratoplasty (PK) replaces the entire thickness of corneal tissue [14]. The advent of endothelial keratoplasty (EK) has evolved the surgical approach to corneal endothelial dystrophies (i.e., Fuch's endothelial dystrophy (FED), congenital hereditary endothelial dystrophy), and other corneal insults causing endothelial pump failure. EK is performed via various techniques, including Descemet's stripping endothelial keratoplasty (DSEK), Descemet's membrane endothelial keratoplasty (DMEK), Descemet's membrane automated endothelial keratoplasty (DMAEK), and posterior lamellar keratoplasty (PLK) [15,16]. Usually, for lamellar keratoplasty and penetrating keratoplasty, donor corneal tissue is secured to recipient corneal tissue via peripheral sutures. Meanwhile, endothelial keratoplasty (EK) has evolved to the use of appositional intracameral gas bubbles and ophthalmic viscoelastic solution.

## 2.2. Presurgical Protocols

Proper patient selection is key to yielding reasonable CT outcomes. A thorough corneal, anterior and gross posterior segment evaluation, in addition to a systemic workup, should ideally be carried out prior to qualifying all potential CT candidates. With regard to preoperative corneal evaluation, the depth of corneal tissue scarring and evidence of corneal ectasia should be determined. Detection of keratic precipitates (mutton fat, stellate or pigmented) indicates active or latent migration and deposition of immune moieties. The resultant hyperimmune sensitization disrupts the ocular immune privilege, decreasing the chances of post-graft tissue survival by more than four-fold [17]. Established intrastromal corneal neovascularization also often portends enhanced migration of phagocytes to host corneal tissue, thus compromising ocular immune privilege [18]. Clinical signs of anterior synechiae should also be looked out for, especially for EK candidates.

When significant corneal stromal haze or widespread leucomatous scarring impairs intraocular evaluation, imaging modalities such as anterior segment-ocular coherence tomography (AS-OCT), ultrasound biomicroscopy (UBM), and B scan ultrasonography can be respectively utilized to gain a gross outline of individualized intraocular integrity. Corneal central thickness (CCT) measurements with pachymetry have, in the past, been correlated with corneal endothelial cell (CEC) loss [19], mostly among contact lens wearers and pseudophakic individuals [20]; hence, it may be an empirical indicator of viable host endothelial physiology prior to lamellar keratoplasty [21]. The patient's innate or adaptive immune status plays a fundamental role in CT outcomes, especially in terms of the risk of post-graft rejection [22].

Under normal circumstances, the trilaminar cornea is devoid of lymphocytes. The cornea's innate defense systems are provided by bacteriostatic enzymes in the tear film, the equilibrium of microbiome/normal flora on the ocular surface and the corneal epithelial barrier attributes [23]. This innate non-infiltrated status of healthy corneal tissue is commonly termed the "ocular immune privilege". In cases of chronic infectious or inflammatory keratitis, however, polymorphonuclear lymphocytes are sensitized to migrate to offended corneal tissue, either through circulation via aqueous humor and/or progression of capillary vascular loops within corneal interstices [24]. These adaptive changes in corneal physiology have been linked with very poor graft survival outcomes post-CT. Host tissue fitting these unwanted criteria is advisably ruled out for allograft CT [25].

Standardized procedures of donor corneal tissue viability analysis are usually followed prior to harvesting and cryopreservation with corneal banks [26]. With regards to recipient immune status, physiological or disease states inducing immune dysregulation [25] such as pregnancy, poorly managed diabetes mellitus (DM), active tuberculosis, an acquired

immunodeficiency syndrome (AIDS) or human immune virus (HIV) with low CD4 counts should be screened out.

Rooij et al. conducted a comparative analysis of three standard methods of CT for the management of Fuchs endothelial dystrophy. They found that there was no significant difference between improvement using conventional PK, inverted mushroom PK and DSAEK [27].

## 2.3. Sources of Cornea Transplant Material

Globally, the need for donor corneal tissue to ease the burden of avoidable corneal blindness far outweighs the availability of viable graft tissue for CT [26]. The demand for viable corneal grafts is further increased by the need for repeat keratoplasty for eyes with a previous history of graft rejection [28]. Although widespread socioeconomic and religious differences contribute to difficulties with harvesting corneas promptly postmortem, other morphological, anatomical and physiological hindrances skew the demand-supply of corneal grafts to the negative [29]. The scarcity of corneal allografts suggests that alternative strategies for obtaining viable cornea transplant material could help close the gap, particularly in developing and underdeveloped areas [30].

Safe strategies for extending the viability period of donor tissue before allografting have been explored [31]. Glycerol-preserved corneas have been deemed comparable to fresh corneal tissue for emergent transplantation needs [32]. Conventionally, penetrating keratoplasty is carried out using allograft [28]. The implantation of "cell-free" biomaterials enabling corneal regeneration may evolve into human applications in the future, particularly for high-risk CT cases. This transplantation strategy has been applied to porcine eyes via anterior lamellar keratoplasty, with good tissue stability but inconsistent visual resolution accounted for [33].

Autologous graft sourcing also represents a viable strategy in rare cases where healthy corneal tissue on the contralateral side may not be of great functionality due to late comorbid ocular disease [34]. Homonymous corneal transplantation has been carried out post-enucleation for a patient with a central leucoma on the right eye and large choroidal melanoma on the left eye; findings thus far suggest that preoperative histopathological assessment of corneal-limbal tissue may be reliable enough to evade chances of neoplastic spread via autografting from such diseased eyes [34].

Corneal transplantation of autologous tragal perichondrium with an amniotic membrane overlay has been reported to yield symptomatic relief among patients with bullous keratopathy compared to controls [35]. Culturing human corneal endothelial cells from mesenchymal stem cells originating from the neural crest has also been explored [36].

## 2.4. Wound Healing after Cornea Transplant—New Perspectives

A common complication of wound healing is neovascularization. Topical bevacizumab application post-CT four times daily for 24 weeks was found to statistically reduce the occurrence of neovascularization [37]. Better results were obtained when bevacizumab was administered subconjunctivally compared to topically [38].

Immune rejection after CT has been successfully managed via donor-derived, tolerogenic dendritic cells [39]. A blockade of E-selectin adhesion receptors [40] has been shown to prolong CT graft survival. The monoclonal antibody RMT1-10 also enhanced the survival of corneal allografts in-situ [41]. Matching donor and recipient human leukocyte antigens (HLA) may also prevent an autoimmune breakdown of the graft after CT [42]. Topical application of cyclosporine nanomicelle eye drops (CNED) may also suppress the immune response by downregulating the NF-κB and ICAM-1 expression [43].

## 2.5. Other Uses of Corneal Transplants

In cases where there is a severe corneal ulceration threatening the integrity of the globe, a penetrating keratoplasty can be the last chance of saving that eye [44]. The key cutoff is when the corneal endothelium is compromised. Therapeutic-tectonic penetrating

keratoplasty continues to show promise in eyes affected by infectious corneal ulcers. Jafarinasab et al. published a review of 32 eyes that had suffered blunt trauma and were at high risk of globe rupture due to corneal perforation secondary to infections. Twenty-six eyes eventually underwent penetrating keratoplasty, and five underwent DALK. One eye was managed with a Tectonic graft [45]. They reported significant improvements in globe stability, although some eyes suffered a recurrence of the presenting infections. Doğan and Arslan reported even better outcomes, with 42 out of 43 patients achieving stable globe integrity [46].

*2.6. Preparation and Storage of Donor Corneas*

Corneal donation involves identifying a potential donor eye, preparing the eye for removal of the graft, extraction of donor tissue and storage. Therefore, the preparation of corneal grafts takes up a critical stage in this regard due to the high demand for cornea tissue and relatively lower supply [47,48]. Intra-operative optical coherence tomography has been useful in the preparations of corneal tissue during extraction. Donor tissue preparation can also be done under this new technology [49]. This technique showed a remarkable 98.5% match with other standard clinical grading. Specular spectroscopy can be used to evaluate endothelial cell density. Modis et al. suggested that specular and non-specular spectroscopy can be used as exchangeable [50]. Bonci et al. compared two methods of preparation of donor corneas; one method used a Hanna trephine with an external chamber, while the other used the same Hanna trephine but with a punch [51]. The methods extracted the donor cornea along the epithelium-endothelium and endothelium-epithelium directions, respectively. The study revealed that endothelial cell density (ECD) loss was less in the epithelium-endothelium cut corneas. As ophthalmic surgeons are ultimately responsible for carrying out these procedures, it is imperative that they keep abreast with different developments in the field by retraining and reviewing relevant literature.

Hagenah and Winter reviewed recent trends in globe disinfection before graft extraction [52]. They suggested a 3% povidone-iodine solution for disinfecting the globes. It is also advised that extracted corneal tissue should be secured in a corneoscleral disc for some time to antibiotics neutralize any remaining contaminants. They recommended that storage containers can be sealed or changed every two weeks under aseptic conditions [52]. Wykrota et al. examined activities at an Eye bank over a period of nine years and reported that low endothelial cell count (ECC) was the most common reason for discarding banked corneas, followed by tissue contamination [53]. Solley et al. reported a new device for preparing grafts via Descemet membrane endothelial keratoplasty in diabetic and non-diabetic eyes. This new device, the DescePrep, performed well with a 97% success rate and average cell death post-extraction at 7.9% $\pm$ 3.7% for all corneas [54].

The ECD can be a very important indicator of the eventual viability of the graft. Gupta and Gupta examined 100 eyes that received corneal transplants within a two-year period over periodic intervals. They reported that the most endothelial cell loss was seen in patients undergoing a repeat graft, while keratoconus-grafted eyes showed the lowest loss [55]. Another study found that while ECD was not related to endothelial decompensation after transplant, it was positively r = correlated with the same at six months post-operation [56]. This study also reported a drop in ECD to lower than 500 cells/mm$^2$ in 14% of its participants and co-morbidities in corneal transplant medicine.

This relative scarcity of corneas for transplant puts their storage on the front burner for many researchers. The numbers of overall grafts properly stored and the eventual viability of the grafts when needed for eventual transplant are two key indices that can advise on the progress of corneal tissue storage development. Garcin et al. developed an active storage device which they called a bioreactor and compared its preservation abilities versus the organ culture storage technique. They published results showing that the bioreactor had a 23% higher rate of endothelial cell survival than the organ culture technique [57]. They also reported better transparency and tissue integrity in the corneas stored using the bioreactor as compared to the organ culture technique. The authors also showed increased longevity

of tissue preservation by using active storage. They conducted a random trial using 24 paired corneas equally split between organ culture and their active storage machine. Their study showed that the active storage system maintained tissue integrity, endothelial cell viability, and numbers in addition to the expression of biochemical stability markers like CD166. In contrast, the organ culture-stored corneas were all found to be unsuitable for corneal transplant at the same time period [58].

Generally, the causes of CT failure have been classified into:

- Failure of the cornea graft (PGF).
- Immunological reaction by the host to the graft [59].
- Non-immunological factors such as spontaneous decompensation, glaucoma and diseases [60,61]. Wound dehiscence and surgical trauma are included in this classification [62].
- Other distinct factors [63].

The cornea endothelium is an inner barrier to edematous damage from aqueous humor breaching into the cornea. Endothelial cell count (CEC) is therefore essential to assess the ability of the eye to retain this protective ability. Studies have shown that the CEC is significantly reduced when corneas are extracted for grafting [64]. Jafarinasab et al. reported on the outcomes of CT in 32 eyes that had previously suffered globe rupture. They reported a progressive reduction in stability in the host-graft interface. They also suggested that an uncompromised Descemet's membrane layer can help to mitigate this occurrence [65]. This Endothelium/Descemet complex thickness (En/DMT) is a good indicator of the prognosis of stability after a successful CT. In fact, an increased En/DMT thickness is an indicator of tissue rejection post-CT, especially in high-risk corneas [66].

Cytokine elevation has also been observed in eyes that suffered CT failure [67], although topical postoperative drugs like levofloxacin, Fk506 and dexamethasone had no deleterious effect when used post-PKP [68,69]. Any disruption in the normal homeostatic and tissue balance in the cornea can first result in corneal haze (CH) [69]. Pentacam Scheimpflug densitometry was used to measure CH in 44 and an incidence of 6.8% (95%CI: 1.4–18.7%) in a tertiary hospital-based study [70]. Corneal densitometry measured $21.86 \pm 6.22$ GSU preoperatively in the 44 eyes. However, this reduced to $21.23 \pm 4.29$ GSU at 12 months postoperatively ($p = 0.815$).

Wound dehiscence is a common complication post-CT. A study was conducted in the Riyal Eye and Ear hospital to assess the incidence and clinical demographic of 72 eyes who visited a center for surgery [71]. Perioperative wound dehiscence may also occur at the host-graft junction and lead to a failure of the CT [72]. The presence of lymphatic vessels within neovascularized failed CT has also been implicated. It remains to be determined if this suggests a new direction for future preventive management [73]. Fine needle diathermy has been indicated as a novel method of managing this lymphangiogenesis [74].

Corneal opacities impede intraoperative visualization during cataract extraction and vitreoretinal surgery [75]. Performing keratoplasty prior to other secondary intraocular surgeries, however, may risk graft tissue viability. Hence, bi-procedural approaches have been explored. Combined keratoplasty and pars plana vitrectomy (PPV) has been successfully performed. The study reported, however, high rates of graft failure, which were correlated with active intraoperative corneal inflammation [76]. Planned EK following cataract surgery has also been recommended amongst patients with endothelial dysfunctions: dystrophic, degenerative or secondary to ocular disease like glaucoma [75]. Features consistent with the ocular dysgenesis disorder, aniridia, have been shown to manifest onto donor corneal buttons following transplantation to diseased corneas [77].

Clinicians should also note that patients receiving immune checkpoint medication are at risk of CT rejection [78]. Common systemic conditions like diabetes can encourage infection and delay wound healing. Infective agents/diseases have also been implicated as sources of poor outcomes. Donors may be the source of infective pathogens that eventually lead to a CT rejection in the future. A Saudi Arabia study found that 0.88% of donated corneas used in 684 CT cases were positive for infective pathogens [79]. Candida infections are commonly reported as culprits in CT failure [80]. In a review of 789 eyes that underwent

CT at two Indian centers, five eyes (0.63%) developed complications secondary to Candida infections. Contributing risk factors were prolonged steroid use, repeated CT, and cornea epithelial weakness [81].

Another study at a different center reported an incidence of microbial keratitis in 4.77% of 1508 eyes that had undergone CT [82]. Natamycin and voriconazole have been suggested as viable therapeutic regimens for managing Candida infections [83]. Mucormycosis has been registered post-deep-anterior-lamellar-keratoplasty (DALK) [84], with a suggestion to manage any sinusitis in a diabetic patient before attempting CT surgery. Giurgola et al. have suggested that Kerasave, a new synthetic product, may drastically reduce in-vitro contamination of donor corneas [85]. Animal models have also indicated that injection of mesenchymal stem cells may limit acute post-CT inflammation [86].

Non-infective unforeseen events can also complicate the CT process. Jarstad et al. reported a case of cardiac failure in an otherwise healthy young adult during CT surgery that was successfully managed [87]. Another common complication of CT is glaucoma [88]. Management may include pharmacological agents or surgery. Studies have shown that surgery for glaucoma after CT has been implicated in CT rejection [89].

Rare conditions like monoclonal gammopathy of undetermined significance are also implicated, especially in the presence of elevated levels of abnormal M proteins [90]. Paniz-Mondolfi et al. also reported the isolation of Colletotrichum chlorophyti in an eye with postoperative endophthalmitis after CT [91]. Another study reported a post-CT infection of Alternaria alternate [92]. The human cytomegalovirus (CMV) is known to attack tissue all over the body. Like all viral conditions, it can lie dormant for extended periods and become active without warning. CMV endothelitis results in loss of CEC and subsequent poor prognosis.

As with any grafting procedure, immunologic interactions between the host and donor tissue can be problematic. However, non-immunologic factors can also be a factor in the loss of CEC, even with patients receiving autologous CT [93]. Repeat transplant/graft procedures are also a risk factor in the failure of CT. Aboshisha et al. conducted a large-scale study to examine the survival rates for regraft procedures for keratoconus, Fuch's endothelial dystrophy, and pseudophakic bullous keratopathy (PBK). PK and EK were the procedures of choice for FE dystrophy and PBK, while PK and DALK were the procedures of choice for keratoconus [94]. Sometimes, the failure is not in the procedure nor the result but in the patient's experiences after CT [95]. Masmoudi et al. reported a patient suffering from Charles Bonnet syndrome after CT [96].

*2.7. The Effect of COVID-19 on Corneal Transplant Medicine and Services*

The avascularity and degree of sensory innervation of the cornea ensure the maintenance of its structural integrity and function. A deviation from the regular structural integrity and outline precipitated by cornea diseases is the fourth leading cause of blindness worldwide. Most visual impairments resulting from cornea morbidities are avertible and reversible. Cornea transplantation is a required surgical intervention in cornea diseases which results in a deep lamellar cornea structural deformity; however, as with most solid-organ transplants, the risk of transmission of infectious diseases to the handlers of donor tissues and recipients cannot be overlooked. The Eye Bank Association of America (EBAA) released updated recommendations and policy changes with the advent of the COVID-19 pandemic [97].

The COVID-19 pandemic resulted in acute and chronic shortages of donor corneas in a healthcare system that already had a long list of patients waiting [98,99]. A study from Riyadh, in which data from patients who underwent keratoplasty between January 2006 to March 2021 were retrospectively reviewed, showed aphakic/pseudophakic bullous keratopathy, keratoconus, regraft, and keratitis as the most common indication for keratoplasty. Furthermore, it was found that lamellar keratoplasty was more favored than penetrating keratoplasty during that era. There was a significant reduction in CT during the COVID-19 period. These researchers reported that painful eyes and emergencies were

prioritized [100]. Due to safety concerns, most surgical procedures without an emergency tag were often delayed or canceled. Reviews of the Italian national eye bank report, in which data from 13 out of 13 Italian eye banks were analyzed, demonstrated a statistically significant decrease in the number of donor corneas retrieved in 2020 as compared to a similar era in 2019 and 2018 [101,102].

Several questions regarding the spread and transmission of SAR-CoV-2 in tears and ocular tissues have been asked. The first was to identify the presence of SAR-CoV-2 in tears which may enable its transmission and subsequent infection of ocular tissues [103]. In a study of 38 confirmed cases, 31.6% of the patients were found to have conjunctival signs, although most of these patients were in the critical phase [104]. Consequently, the screening of donor corneas from COVID-19 patients is of great relevance. The insufficiency of evidence-based studies establishing the presence of the COVID-19 virus in the cornea and tears of COVID-19 patients, however, has delayed evidence-based consensus. As such, preferred practice guidelines ensure zero risk of transmission of the disease be instituted [105]. The short-term availability of transplant corneas after harvesting makes cornea transplant an always urgent surgical intervention; consequently, satisfactory safety adjustments to eye clinics and ophthalmic surgery organizations, as well as reliable diagnostic procedures, should be favored to minimize the risk of transmission of SARs-CoV-2 [106].

The shortage of donor tissue during the COVID-19 pandemic also had positive results. Clinicians reported an innovative procedure and were able to use a single donor cornea for four different recipients undergoing CT for different conditions [107]. Another study reported three patients benefitting from a single cornea [108]. Access of CT recipients to needed care also became constrained during the early pandemic months [109]. Procurement and distribution channels for donor corneas were also disrupted worldwide due to flight restrictions and lockdowns [107,110]. Joshi et al. reported on 10 patients who suffered CT failure in an Indian setting. They opined that the fear of contracting SARs-CoV-2 was a factor that resulted in the delayed presentation of these patients [111]. Thuret et al. assessed the effect of the COVID-19 pandemic on corneal transplants in Europe. In particular, they looked at how corneas were obtained and distributed to eye banks between May 2018 and 2020. They concluded that the additional screening and testing protocols occasioned by the COVID-19 pandemic resulted in a decrease in the donor pool, as expected. They recommended that evidence-based protocols and recommendations should be investigated to ensure that the new screening and testing protocols are justified [112].

### 2.8. Cornea Transplant in Developing Countries

A 15-year review of corneal donations in southern Africa revealed that cultural beliefs have led to a marked reduction in donation numbers [113]. However, another study among drivers in Africa reported that 67.3% of participants were willing to sign up to donate their corneas after obtaining some knowledge of the importance of corneal transplants. Eze et al. carried out a questionnaire to assess the knowledge and attitudes in a University environment [114] in Nigeria. The responses of clinical-based students were compared to non-clinical students. They found significant deficits in knowledge and attitudes to ocular tissue transplant and donation. They suggested that the introduction of the subject matter of corneal transplants into the undergraduate coursework and awareness will improve understanding and knowledge among the two groups. This was also corroborated by work credited to Williams and Muir [115]. The William and Muir study suggested factors including age, race and being Muslim was correlated with a higher willingness to donate corneas. However, they reported that a modifiable factor that affected willingness to operate was knowledge and awareness. In fact, Wang et al. suggested that patients who had required corneal transplants in the past were more likely to show better knowledge and accede to donating their corneas [116,117]. Hussen et al. suggested implementing awareness creation programs in communities [118].

Gain et al. conducted a systematic review of reports on corneal transplant medicine and eye banks. They then conducted interview series among specialists spanning 148

different countries. In total, they reviewed 184,576 surgeries using 283,530 corneas obtained from 742 cornea banks. Their work concluded that there was indeed a scarcity of corneas worldwide. Their calculated availability was estimated at only one cornea for every 70 eligible transplant patients. They suggested an acceleration of alternative/artificial solutions such as keratoprostheses [119].

*2.9. Current Perspectives on CT*

The first CT took place in 1884 using a bovine cornea [120]. CT has come a long way since then. Recent publications suggest partial thickness CTs are the most common procedures performed in eye centers, with FED being the leading cause of corneal disease both in the United Kingdom, Hong Kong and Canada [121–125] as compared to previous decades when full thickness CT was common. The opposite was reported in Poland, where full-thickness CT made up 82% of surgeries in a study [125].

Amagai et al. reported the successful development of a novel tissue carrier for donor graft tissue [126]. They validated a poly(lactic acid)-based carrier for cell sheets transplantation of cornea sheets in animal models. All-laser CT procedures have been reported to improve outcomes [127]. Femtosecond Laser-assisted cornea matrix transplantation has also shown promise in correcting ametropia [126,128].

Tissue-engineered construct models for corneal endothelial keratoplasty have shown greater pharmacoeconomic potentials than those working with donor tissue [127]. Ultraviolet corneal cross-linking yields greater than 80% corneal topographic stability postoperatively. However, PK and DALK remain first-line surgical options for advanced keratoconus (KC), with a significant complication profile. Thus, Bowman's layer transplantation is a promising strategy for avoiding very invasive keratoplasty at advanced stages of KC [129]. Intraoperative AS-OCT has been reported to be of great value during PLK in the case of congenital posterior polymorphous dystrophy [130]. For determining corneal endothelial cell density (ECD) post-grafting, confocal microscopic 'polygonal' variable cell count has been deemed the most clinically applicable method, with comparable reliability to planimetry [131].

Azithromycin treatment regimen has been reported to reduce the incidence of high-risk allograft rejection in murine models [132]. Preoperative incubation of donor corneas in media containing recipient-specific human serum (HS) has been suggested to yield improved graft success outcomes in the eighth year post-CT [131]. It has been recommended that performing limbal stem cell allograft transplantation from human leukocyte antigen (HLA)-matched donors, combined with amniotic membrane transplantation prior to PK-induced graft success outcomes in a patient with a previous history of multi-graft rejection [133]. Some promise has been depicted via angiopoietin-2 blockade therapy post-allogeneic corneal transplantation by inhibiting lymphangiogenesis [134].

Steroid therapy after ocular surgery is indicated for the prevention of inflammation and also for its immunosuppressive properties. Poinard et al. conducted a study to assess the adherence of cornea transplant receivers to prescribed postoperative steroid therapy [135]. Steroid therapy is usually maintained topically for one-year post-transplant. They monitored a cohort of 30 patients for a duration of one year after surgery using a KaliJAR device that recorded the number of doses of eyedrops leaving a vial. They observed a high correlation between the KaliJAR-reported and self-reported doses of steroids in the patients. They, therefore, concluded that the KaliJAR was an accurate device that has potential applications in monitoring patients with poor drug adherence.

Trone et al. also conducted a pilot study (Phase II) as part of a clinical trial to assess the safety of a new dropless PK which incorporated a subconjunctival steroid implant [136]. Fourteen patients received this implant after a corneal transplant and were monitored for changes in IOP and corneal transplant stability indices. Indices included inflammation, central corneal thickness, and tissue rejection. They successfully achieved stable IOP in all participants and reported an average resorption period of 1.5 months, with one patient suffering a rejection.

*2.10. Keratoprostheses: A Game Changer*

As already stated, conventional keratoplasty involves allogeneic transplantation of corneal tissue: tissue-specifics depending on the disease section of the cornea [137]. Notably, penetrating keratoplasty (PK) is preserved for severe inflammatory, infectious or degenerative keratitis or resolution of visual impairment with associated extensive corneal scarring [138]. Limitations to PK for such indications include immune responses to donor antigens and poor supply to meet the demands for viable trilamellar corneal tissue [139]. The advent of artificial corneal transplant material, also termed keratoprosthesis (KPro), may well provide long-term solutions to pressing questions probing the field of corneal preservation and transplantation [140].

Keratoprestheses require a clear central optic zone to enable refraction onto the retina. Historically, early KPro material consisted of convex glass sheets which served as optics surrounded by metallic rims made of silver, quartz, platinum and like materials [141]. Attempts were also made to secure corneal tissue between plates of glass. The era of KPros gained greater validity and application following the development of polymethylmethacrylate (PMMA)-based biologically integrable material for the replacement of failed allogeneic grafts, as well as the early evolution of surgical applications.

Retrospectively, the Boston type-1 KPro (with its phakic and aphakic models) and the Osteo-odonto KPro have reportedly recorded proven highest success rates. The use of the temporary Eckardt KPro and other retrievable soft contact lens media has been successfully trialed for posterior segment visualization during vitreoretinal procedures with combined keratoplasty [142,143]. Eyes with good tear secretion and non-severe ocular surface disease (OSD) are recommended to fair better with the Boston keratoprosthesis, while on the other hand, cases of marked OSD with severe structural adnexal defects are suggested for the use of Osteo-odonto KPro.

Preoperative preparation requires ocular imaging and biometric analysis to determine anterior chamber depth; gross morphological iridocorneal angle integrity; crystalline lens presence/position; also, and retinal function and integrity (which may require electrophysiology in doubtful cases).

Major complications reported post-KPro implantation include glaucoma, persistent autoimmune reaction with cases of coexisting system autoimmune disease (leading to melting and chronic inflammation, infection, and retinal detachment. However, it is worth stating that proper preoperative or perioperative identification of ocular surface or related human leukocyte antigen (HLA)-driven responses can help to avert further sight loss following KPro implantation.

## 3. Conclusions

Corneal specialist surgeons should be globally incentivized to encourage uptake, training, and retraining in modern techniques of allograft transplantation. The major impediment to optimal management of corneal blindness aside from donor shortage remains post-CT rejection/graft failure. Finding solutions may depend on the rate of progress regarding the incorporation of either biological material or procedural modulation of stem cell differentiation to avascular corneal-like tissue, especially for immune-mediated graft failure. There will certainly be limitations to this approach, specifically in the exploitation of totipotent embryonic stem cells. Strategies to prolong the viability time of donor grafts would also serve to minimize cost effectiveness preoperatively. Demand for CT may also decrease if current advances in local ocular stem cell application and amniotic membrane transplantation reduce the incidence of corneal scarring following chemical injury and deep corneal ulceration.

**Author Contributions:** Conceptualization, M.M. and M.Z.; methodology, M.M., M.Z., E.C. and E.S.E.; validation, M.M., M.Z. and C.S.; formal analysis, M.M., M.Z., E.S.E. and E.C.; investigation, M.M., E.C. and E.S.E.; resources, C.S.; writing—original draft preparation, M.M. and M.Z.; writing—review and editing, M.M., E.S.E., E.C. and M.Z.; visualization, M.M., M.Z., E.S.E., E.C. and C.S. supervision, C.S.; project administration, C.S. All authors have read and agreed to the published version of the manuscript.

**Funding:** This research received no external funding.

**Institutional Review Board Statement:** Not applicable.

**Informed Consent Statement:** Not applicable.

**Data Availability Statement:** Not applicable.

**Acknowledgments:** The authors wish to acknowledge Godwin Okoye of Africa Eye Laser Centre Ltd. and Centre for Sight Ltd.

**Conflicts of Interest:** The authors declare no conflict of interest.

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
