# Peer review of "An Overview of Corneal Transplantation in the Past Decade"

_clinpract, doi:10.3390/clinpract13010024_

Round 1
Reviewer 1 Report
Congratulations for authors on this great effort to present an overview of corneal transplantation in the last decade.
The objective of this journal is to present review articles in clinical medicine in general. So this review should tackle different angles of corneal transplantation.
Authors describe corneal transplantation for optical purposes. It will be very useful for readers to explain that corneal transplantation can also help to restore globe integrity in those cases with imminent risk of corneal perforation, for example in bacterial keratitis (therapeutic-tectonic penetrating keratoplasty).
Amniotic membrane transplantation should not be part of this review article; instead of this, I would recommend authors to mention how corneas should be harvested and properly preserved in an Eye Bank before corneal transplantation. Preoperative donor corneal tissue evaluation should also be described in detail because it is the utmost importance to measure endothelial cell density before deciding to use it and also to rule out viral and bacterial infections in the donor that can be potentially transmitted through corneal tissue to recipients .
Eye donation programs and how to increase awareness of general population -especially in non industrialized countries- were not mentioned.
Artificial cornea development (keratoprosthesis) should also be an important part of discussion of this article because it can be a therapeutic option when human corneas are not available.
Addition of more tables or figures could help readers to understand indications of corneal transplantation, surgical techniques, management of graft rejection and postoperative complications.
References should be written according instruction for authors (following this journal citation style)
Author Response
Please see the letter attached.

Reviewer 2 Report
This paper, which summarises 246 articles on corneal transplantation from 2012 to 2023, is clear and well written. It has the advantage of summarising in a succinct manner (in a few pages) what is involved in this field.
It takes up in several points, the path of the graft from the source to the surgical technique by defining the different techniques used (transfixing, partial, endothelial grafts...) and by highlighting the difficulties of the moment (covid 19) and the perspectives of these grafts. Nevertheless, it seems to me that there is a lack of many references for a synthesis on corneal transplantation and what surrounds it.
I am nevertheless surprised that this summary does not include articles by the team of Philippe Gain and Gilles Thuret (Saint-Etienne, France). This team published an article on the supply and demand for corneal transplants in the World
[Gain P, Jullienne R, He Z, Aldossary M, Acquart S, Cognasse F, Thuret G. Global Survey of Corneal Transplantation and Eye Banking. JAMA Ophthalmol. 2016 Feb;134(2):167-73. doi: 10.1001/jamaophthalmol.2015.4776. PMID: 26633035.].
In addition, they have been working for several years to improve the conservation of banked corneas with the invention of an active corneal bioreactor for eyebanking which significantly reduces the loss of DCE during storage (in the process of being industrialised).
[Garcin T, Gauthier AS, Crouzet E, He Z, Herbepin P, Perrache C, Acquart S, Cognasse F, Forest F, Thuret G, Gain P. Innovative corneal active storage machine for long-term eye banking. Am J Transplant. 2019 Jun;19(6):1641-1651. doi: 10.1111/ajt.15238. Epub 2019 Jan 25. PMID: 30589181.].
They have also worked on the treatment of graft rejection (on rabbits and humans) by subconjunctival implants in order to alleviate the compliance problems of transplanted patients
[Trone MC, Poinard S, Crouzet E, Garcin T, Mentek M, Forest F, Matray M, Thuret G, Gain P. Dropless penetrating keratoplasty using a subconjunctival dexamethasone implant: safety pilot study. Br J Ophthalmol. 2021 Aug 23:bjophthalmol-2021-319376. doi: 10.1136/bjophthalmol-2021-319376. Epub ahead of print. PMID: 34426402.].
Adherence to corticosteroid therapy after transfixing keratoplasty itself has been measured by the same team
[Poinard S, Garcin T, Trone MC, Mentek M, Lambert C, Bonjean P, Renault D, Thuret G, Gain P, Gauthier AS. Objective measurement of adherence to topical steroid medication after penetrating keratoplasty using an electronic monitoring aid: A pilot study. Digit Health. 2022 Sep 16;8:20552076221121155. doi: 10.1177/20552076221121155. PMID: 36133001; PMCID: PMC9483967.].
[Garcin T, Gauthier AS, Crouzet E, He Z, Herbepin P, Perrache C, Acquart S, Cognasse F, Forest F, Gain P, Thuret G. Three-month Storage of Human Corneas in an Active Storage Machine. Transplantation. 2020 Jun;104(6):1159-1165. doi: 10.1097/TP.0000000000003109. PMID: 31895867.]
I think it would be important to take an interest in their work and to cite them in view of the work they have done and continue to do on corneal transplantation.
They also worked on the impact of covid 19 on corneal transplantation in Europe.
[Thuret G, Courrier E, Poinard S, Gain P, Baud'Huin M, Martinache I, Cursiefen C, Maier P, Hjortdal J, Sanchez Ibanez J, Ponzin D, Ferrari S, Jones G, Griffoni C, Rooney P, Bennett K, Armitage WJ, Figueiredo F, Nuijts R, Dickman M. One threat, different answers: the impact of COVID-19 pandemic on cornea donation and donor selection across Europe. Br J Ophthalmol. 2022 Mar;106(3):312-318. doi: 10.1136/bjophthalmol-2020-317938. Epub 2020 Nov 26. PMID: 33243832.]
In view of this surprising lack of referencing, it is possible that your inclusion criteria have significantly restricted the articles you have considered. Have you considered articles containing the words "Transplantation"; "Keratoplasty" in addition to "Transplant"? In my opinion, sorting using only the word "Transplant" has meant that you have missed a lot of work.
I recommend that you at least look at the work of this team (Philippe Gain and Gilles Thuret) to complete your synthesis.
Author Response
Please see the letter attached.

Round 2
Reviewer 1 Report
Congratulations on your corrections !
Your paper need a little minor revision
(typing minor mistakes in Figure 1, some redundant words in additional text. For example: "...revealed that cultural beliefs among other beliefs...")
Reviewer 2 Report
This review, well written by its authors, gives in a few pages a state of the art of corneal transplantation from graft preparation to the treatment of graft rejection by steroids.
The authors also summarise the indications for the conservation of these tissues and the various grafting techniques.
The strength of this paper is the amount of information presented in a few pages, which may be of considerable interest to the scientific community, and which covers all the stages of corneal transplantation. Moreover, the review opens up to new perspectives of keratoprosthesis, to the different techniques of conservation....
It would be very interesting to add statistical data on the different grafting techniques used in the world, such as the percentages of DSEAK, DMEK, KP indications.... What do you think about it?
Like any state of the art, the authors had to limit themselves in the quantity of publications and themes for this paper, making this work perfectible, but it remains highly interesting for the neophyte but also for the expert/surgeon who would need a refresher.
